# Cross-Viewpoint Semantic Mapping: Integrating Human and Robot Perspectives for Improved 3D Semantic Reconstruction

**DOI:** 10.3390/s23115126

**Published:** 2023-05-27

**Authors:** László Kopácsi, Benjámin Baffy, Gábor Baranyi, Joul Skaf, Gábor Sörös, Szilvia Szeier, András Lőrincz, Daniel Sonntag

**Affiliations:** 1Department of Interactive Machine Learning, German Research Center for Artificial Intelligence (DFKI), 66123 Saarbrücken, Germany; 2Department of Artificial Intelligence, Eötvös Loránd University, 1053 Budapest, Hungary; 3Nokia Bell Labs, 1083 Budapest, Hungary; 4Department of Applied Artificial Intelligence, University of Oldenburg, 26129 Oldenburg, Germany

**Keywords:** 3D semantic maps, semantic matching, superpixel segmentation, semantic segmentation, human–robot collaboration, real-time reconstruction, label transfer, computer vision, deep learning

## Abstract

Allocentric semantic 3D maps are highly useful for a variety of human–machine interaction related tasks since egocentric viewpoints can be derived by the machine for the human partner. Class labels and map interpretations, however, may differ or could be missing for the participants due to the different perspectives. Particularly, when considering the viewpoint of a small robot, which significantly differs from the viewpoint of a human. In order to overcome this issue, and to establish common ground, we extend an existing real-time 3D semantic reconstruction pipeline with semantic matching across human and robot viewpoints. We use deep recognition networks, which usually perform well from higher (i.e., human) viewpoints but are inferior from lower viewpoints, such as that of a small robot. We propose several approaches for acquiring semantic labels for images taken from unusual perspectives. We start with a partial 3D semantic reconstruction from the human perspective that we transfer and adapt to the small robot’s perspective using superpixel segmentation and the geometry of the surroundings. The quality of the reconstruction is evaluated in the Habitat simulator and a real environment using a robot car with an RGBD camera. We show that the proposed approach provides high-quality semantic segmentation from the robot’s perspective, with accuracy comparable to the original one. In addition, we exploit the gained information and improve the recognition performance of the deep network for the lower viewpoints and show that the small robot alone is capable of generating high-quality semantic maps for the human partner. The computations are close to real-time, so the approach enables interactive applications.

## 1. Introduction

Human–robot collaboration opens novel application avenues, as super-human sensing, computation speed, additional degrees of freedom in control, and strength can extend human capabilities in problem solving. Taking the example of healthcare applications [1], there are several challenges to be addressed. Robots must adapt to the unique challenges and constraints of the environment while providing safe and effective care to patients. They need to navigate confined spaces, account for different types of furniture, and avoid obstacles while carrying out their tasks. Moreover, privacy concerns and customization of care plans for individual patients are crucial factors to be addressed.

In physical rehabilitation scenarios [2] for example, a ground robot needs to guide the patient to the optimal place and optimal direction, observe the patient while performing physical exercises, track the exercise in 3D, and provide feedback in case of error. Time constraints for correcting instructions can be demanding, and verbal communication may be necessary. Context-aware guidance requires the recognition of the objects, the reconstruction of the geometry of the space, path planning, and execution. Guidance requires verbal instructions in the patient’s egocentric coordinates. Therefore, complete 3D semantic reconstruction is necessary for the recognition of the spatial relationships between objects, the neighboring relations and free spaces between them, and the transformation to the egocentric coordinate system.

While small robots have several advantages in home scenarios, they also face challenges set by computational constraints and by their form factor. An important practical problem is that small robots have difficulties in recognizing objects viewed from their own unusual perspectives, which are typically not present in their object training database. Humans see objects from the top while ground robots see things from the bottom. Moreover, training datasets such as [3,4,5] are usually annotated from human perspectives, which poses a common challenge in image segmentation. Models tend to generalize poorly to different perspectives [6], resulting in significant segmentation accuracy drops, as shown in Figure 1. To address this issue, one option is to request semantic information from a human partner, e.g., through verbal communication. However, this may not be practical or effective in real-time scenarios. Another option is to use semantic scene completion [7] to infer labels of unknown regions, but these approaches are computationally intensive and may not be suitable for real-time operation on typical robots. An attractive option is to use a semantic map of a higher viewpoint to estimate semantics from a lower viewpoint, as also shown in [8]. This approach can provide the object recognition system with low-angle training samples to improve its performance while preserving multi-view consistency. Furthermore, after deploying the semantic segmentation network, continual learning approaches [9,10] could be employed to update the model on previously unseen data.

We propose ways to generate semantic segmentations from lower viewpoints. Specifically, we transfer semantic labels to various 2D viewpoints from an existing 3D semantic reconstruction using various projection and matching mechanisms, building upon the 3D semantic reconstruction works of Rosinol [11] and Rozenberszki [8]. We study the effects of changing the camera heights and compare different segmentation models. We evaluate the proposed label projection approaches on synthetic data coming from the Habitat [12] simulation framework and on a real-world dataset. We demonstrate that our method can provide semantic segmentation from lower viewpoints with similar accuracy as the 3D semantic reconstruction from the upper viewpoint, and it can be used to train semantic segmentation models on unusual perspectives. Our implementation is available at https://github.com/szilviaszeier/semantic_matching (accessed on 23 May 2023).

## 2. Related Works

Robotic systems are increasingly assisting us in our daily lives. Initially, these agents were hard-coded to complete specific tasks without generalizing capabilities. However, in recent years, greater research emphasis has been put on their autonomy and spatial awareness. Spatial artificial intelligence (SAI) can be divided into four distinct but interdependent layers with increasing complexity: spatial perception, pose tracking, geometry understanding, and semantic understanding. Semantic environment reconstruction is the core of our work. In this section, we briefly discuss relevant reconstruction methods and then introduce the problem of segmentation failure from unusual perspectives, such as those of small ground robots and flying drones.

Three-dimensional semantic reconstruction of indoor environments refers to the task of recreating the geometry and appearance of our surroundings. The reconstruction can be performed using different techniques. Some utilize a LiDAR scanner and segment the 3D point cloud, others reconstruct the environment from 2D segmented images with a structure-from-motion algorithm [13], while others traverse the area with an RGBD or stereo camera and create a map on the fly. In our work, we focus on RGBD cameras.

Due to the complexity and high dimensionality of this task, many solutions only work offline [14,15]. In robotic applications, real-time approaches such as BundleFusion [16] become much more relevant. For example, in a navigation task the robot needs to be aware of immediate changes in its environment to be able to generate a collision-free trajectory [17]. A 3D semantic reconstruction stores environmental information which can later be utilized by the same or other agents to complete various tasks.

Semantic segmentation is a key aspect of the way in which the semantic labels of the reconstruction are created. Some methods utilize 2D semantic segmentation algorithms [18], while others perform the segmentation on the 3D geometry itself [19,20]. There are many publicly available 2D semantic segmentation methods, making this approach an attractive choice. On the other hand, 3D segmentation methods have the advantage of being able to utilize the geometry of a given object, but these techniques are more time-consuming and much fewer variants exist.

In this work, we use 2D semantic segmentation methods [21,22] due to their ease of use and inference speed. Most semantic segmentation algorithms are trained on datasets such as SUNRGBD [3], NYUv2 [4], and COCO [5], which mainly contain images taken from a standard (human) perspective. Consequently, detection is unreliable or fails when applied to images taken from unusual viewpoints, such as those of small ground robots or drones.

Regarding 3D semantic reconstruction, typical methods isolate components and fuse the results later in a pipeline architecture. However, this is not always the case. In [23], the authors propose an approach to jointly infer 3D geometry and 3D semantic labels using a depth fusion network. This method leverages the 2D semantic prior to enhance the 3D reconstruction accuracy, meaning that out-of-distribution viewpoints would not be supported, which is the primary problem we aim to address.

The problem of segmentation failures due to unusual perspectives has been tackled by leveraging multi-view consistency, domain adaptation, and continual learning approaches. A key idea is to exploit the consistency of semantic information across different viewpoints to improve the performance of segmentation networks. The work by [24] highlights the benefits of leveraging view consistency enforced in 3D as a prior for 2D tasks involving semantics. Frey et al. [9] proposes an online adaptation method for semantic segmentation networks during deployment, using a probabilistic accumulation of consecutive 2D semantic predictions in a volumetric 3D map as a supervision signal. This method enforces multi-view consistency through the 3D representation and employs a continual learning experience replay strategy to retain previously learned knowledge. Similarly, Liu et al. [10] introduces a method for continual multi-scene adaptation for semantic segmentation without ground truth labels, using a Semantic-NeRF network for each scene to fuse the predictions of a segmentation model. The Semantic-NeRF model enables 2D–3D knowledge transfer, and its compact size allows for long-term memory storage and reduced forgetting.

These approaches provide ways to adapt the segmentation model to new data in an online or offline manner after deployment. These methods could also benefit from the proposed label projection methods to render 2D pseudo-labels from upper-view 3D semantic reconstructions. We note that segmentation methods keep improving with the increasing size of databases, see, e.g., the recent publication of Kirillov et al. [25]. Nonetheless, the relevance of the problem remains for complex novel objects subject to heavy occlusions, especially in critical applications.

## 3. Methods

In this paper, we contribute several new methods for 3D semantic label transfer and matching across multiple viewpoints.

### 3.1. Reconstruction Pipeline Overview

We first describe the overall structure of our real-time 3D semantic reconstruction pipeline, which is adopted from previous work [8]. The reconstruction pipeline includes UcoSLAM [26] as a visual SLAM method used for pose tracking, and Mask R-CNN [21] as a 2D semantic segmentation algorithm trained on the SUNRGBD dataset [3]. A pose filtering procedure was also introduced to filter out erroneous pose estimates that may result from false re-localization or drifts in the pose. The final 3D model of the environment is constructed using the Voxblox [27] framework by integrating measurements from each sensor into a global map. The reconstruction can be colored or semantically labeled, depending on the inputs. As we focus on 3D semantic reconstructions, we paint the model with semantic labels.

We extended the pipeline of [8] with several new components. The proposed architecture is shown in Figure 2. Our goal is to obtain a 3D semantic reconstruction from unusual viewpoints starting from a partial human-perspective reconstruction. We built a small ground robot to capture data from an odd perspective. The starting reconstruction is provided by the reconstruction pipeline using a typical 2D semantic segmentation algorithm. Originally, Mask R-CNN was used for semantic labeling, but we also included the recent segmentation algorithm called Mask Transfiner [22]. We trained this model on the SUNRGBD [3] and the ADE20K [28] datasets. For comparability, the Mask R-CNN was also trained and evaluated on the same datasets.

Our goal is to extend and improve the starting reconstruction model to unusual perspectives. However, the semantic segmentation algorithms are not reliable from these odd perspectives. Therefore, we propose several different label projection methods which map semantic labels from the starting reconstruction to the perspective of the ground robot.

### 3.2. Label Projection Methods

The label projection module aims to provide 2D semantic segmentation to the point cloud generation module. However, when the camera is close to the floor, the accuracy of the pre-trained semantic segmentation degrades. Therefore, we propose methods to provide high-quality semantic segmentation when the camera position is not optimal by incorporating labels from an existing 3D semantic mesh. In the following sections, we propose five label projection methods: (1) superpixel-based projection in 2D, (2) superpixel-based projection in 3D, (3) supervoxel-based projection, (4) 3D clustering-based projection, and (5) per-point matching.

#### 3.2.1. Superpixel-Based Projection in 2D

To obtain semantically meaningful regions in the RGB image, we utilize superpixel segmentation. Superpixel segmentation is a clustering algorithm that aims to cluster regions of an image based on some similarity metrics such as color, texture, and proximity. We use Fast-SLIC [29], an optimized version of SLIC [30] (simple linear iterative clustering) for superpixel segmentation on CPU-constrained devices.

The superpixel-based projection approach consists of the following steps: (i) We project the 3D semantic reconstruction from the upper viewpoint onto the lower-view image plane. (ii) Fast-SLIC is applied on the RGB frame from the lower viewpoint to group coherent image regions, and the projected semantic labels are aggregated accordingly. (iii) A majority voting scheme is employed to assign semantic labels to each superpixel. This approach is referred to as “2D SLIC”.

#### 3.2.2. Superpixel-Based Projection in 3D

This approach builds on the 2D SLIC method by extending it to 3D. We first perform superpixel segmentation on the lower-view RGB frame using Fast-SLIC as described in the previous section. Next, instead of assigning semantic labels to each 2D superpixel, we project each superpixel onto the lower-view 3D point cloud by using the depth image. We then group the 3D points belonging to each superpixel and assign a semantic label to the group based on the majority of labels. Algorithm 1 provides a detailed description of the method. For the remainder of the paper, we refer to this method as “3D projected SP”.   
**Algorithm 1:** Superpixel-based projection in 3D**Input:** Lower-view RGBD image ($I\in {\left[0,255\right]}^{H\times W\times 3}$ and $D\in {\left[0,255\right]}^{H\times W}$), upper-view 3D semantic reconstruction (including point cloud $P_{upper}\in R^{N\times 3}$ and labels $L_{upper}\in {\left\{0,1,\cdots ,L\right\}}^{N}$, where L∈N denotes the number of semantic labels).**Output:** Lower-view semantic segmentation ($S\in {\left\{0,1,\cdots ,L\right\}}^{H\times W}$)1Run superpixel segmentation (SLIC) on the RGB image (*I*) to find coherent image regions (SP).2Using the depth image (*D*), project the superpixel segmentation (SP) onto 3D to obtain the lower-view point cloud ($P_{lower}$).
3$\left[\mathrm{Optional}\right]$ Downsample the point cloud ($P_{lower}$).4Match the lower-view point cloud ($P_{lower}$) with the upper-view 3D semantic reconstruction ($P_{upper},L_{upper}$) and determine the corresponding semantic label of each superpixel ($L_{lower}$).5Project the labels of the lower-view semantic point cloud ($L_{lower}$) onto the image plane to obtain the semantic segmentation (*S*).

#### 3.2.3. Supervoxel-Based Projection

In addition to the generalization of the label assignment, one can extend superpixel segmentation to 3D. In this case, we refer to the resulting 3D regions as supervoxels. Out of the several approaches for supervoxel segmentation, we use an extension of SLIC called maskSLIC [31].

To apply maskSLIC, we need to convert the 3D point cloud $P\in R^{N\times 3}$ to a voxel grid using the formula $V=\frac{P-\bar{P}}{V}$, where $\bar{P}=\frac{1}{N}\sum_{j=1}^{N}P^{\left(j\right)}$ is the center of the point cloud and V∈R is the voxel size, which represents the length of the side of a cubic voxel in meters. In order to keep the runtime low, we introduce heavy subsampling on the point cloud before the voxelization. Furthermore, we use maskSLIC with a foreground mask to ignore empty voxels, and we limit the number of supervoxels. Algorithm 2 describes the supervoxel-based projection method in detail. We will refer to this approach as the “3D SLIC” method in the remainder of this paper.   
**Algorithm 2:** Supervoxel-based projection**Input:** Lower-view RGBD image ($I\in {\left[0,255\right]}^{H\times W\times 3}$ and $D\in {\left[0,255\right]}^{H\times W}$), upper-view 3D semantic reconstruction (including point cloud $P_{upper}\in R^{N\times 3}$ and labels $L_{upper}\in {\left\{0,1,\cdots ,L\right\}}^{N}$, where L∈N denotes the number of semantic labels).**Output:** Lower-view semantic segmentation ($S\in {\left\{0,1,\cdots ,L\right\}}^{H\times W}$)1Create a colored 3D point cloud ($P_{lower}$) from the lower-view RGB (*I*) and depth image (*D*).2Downsample the point cloud ($P_{lower}$) and keep track of each point’s original location.3Convert the lower-view point cloud ($P_{lower}$) to a voxel grid ($V_{lower}$).4Utilize SLIC on the voxel grid ($V_{lower}$), using masking to obtain supervoxels (SV).5Match the lower-view point cloud ($P_{lower}$) with the upper-view 3D semantic reconstruction ($P_{upper},L_{upper}$), determining the semantic label ($L_{lower}$) of each supervoxel (SV).6Project the semantic labels of the lower-view point cloud ($L_{lower}$) onto the image plane to obtain the semantic segmentation (*S*).

#### 3.2.4. Three-dimensional Clustering-Based Projection

We propose an alternative clustering approach that first identifies planes (e.g., floor, ceiling, and walls) in the scene and then clusters the remaining points. We use the RANSAC [32] algorithm to identify the dominant planes in the point cloud, and the density-based spatial clustering of applications with noise (DBSCAN) [33] algorithm for clustering. Algorithm 3 details the proposed approach.

This approach can handle point clouds with hundreds of thousands of points. However, subsampling the point cloud can improve the runtime. The number of supervoxels cannot be adjusted directly, as it depends on the point density, therefore, it may struggle to distinguish regions with fine details. Furthermore, fine-tuning the parameters may be necessary to reduce sensitivity to noise. For the rest of the paper, we refer to this approach as “DBSCAN”.   
**Algorithm 3:** Three-dimensional clustering-based projection
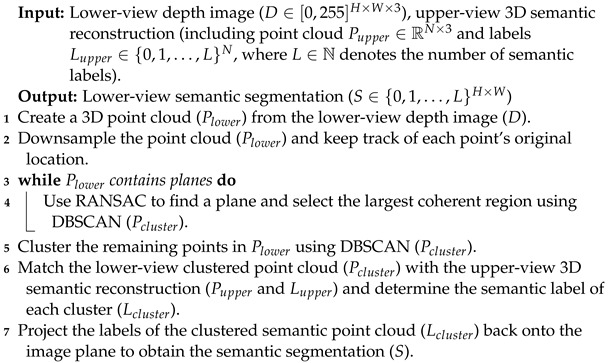


#### 3.2.5. Per-Point Matching

The per-point matching method involves projecting the lower-view RGBD frame to 3D and then comparing the resulting point cloud with the upper-view point cloud to assign corresponding upper-view labels to the lower-view points.

Let $P_{lower}\in R^{M\times 3}$ be the lower-view point cloud, and $P_{upper}\in R^{N\times 3}$ be the upper-view point cloud, and $L_{upper}\in {\left\{0,1,\cdots ,L\right\}}^{N}$ be the corresponding upper-view labels, where L∈N. We assign the nearest upper-view label to each lower-view point:$$L_{lower}^{\left(j\right)}:=L_{upper}^{\left(i\right)},$$
where $L_{lower}\in {\left\{0,1,\cdots ,L\right\}}^{M}$ and ∥.∥ denote the labels of the lower-view point cloud and the $L_{2}$ norm, respectively, and i∈[1,N],j,k∈[1,M],j≠k are indices, such that
$${∥P_{lower}^{\left(i\right)}-P_{lower}^{\left(j\right)}∥}_{2}\leq {∥P_{lower}^{\left(i\right)}-P_{lower}^{\left(k\right)}∥}_{2}\;\;\;\;\;\;\left(\forall k\right).$$

After assigning labels to the lower-view points, we project $L_{lower}$ onto the image plane to obtain the semantic segmentation. We refer to this approach as the “3D neighborhood” method. Note that this method is computationally efficient but can result in errors when there are significant occlusions or changes in viewpoint between the upper- and lower-view frames.

### 3.3. Superpixel-Based Edge Refinement

To ensure accurate instance or semantic segmentation, it is important that the objects’ regions follow a set of pre-defined concepts, such as smooth and spatially precise boundaries between segments [34]. However, the proposed label projection methods may result in rough and uneven boundaries between objects, hindering the fine-tuning of segmentation models on the generated samples. To address this issue, we introduce a superpixel-based pre-processing step using Fast-SLIC [29], as described in Algorithm 4. This method aggregates pixels into superpixels, resulting in smoother boundaries and mitigating the negative impact of ragged edges. Further improvements could be made through post-processing techniques, as described in [35], and by exploiting the 3D nature of scenes through radiance field propagation and bidirectional photometric loss, as discussed in [36].   
**Algorithm 4:** Superpixel-based edge refinement
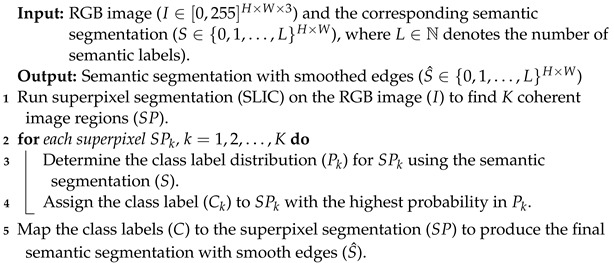


### 3.4. Benchmark Measures

When comparing meshes, we transform them into semantic voxel grids and then compare these grids with the following specified metrics:Jaccard: The Jaccard index is calculated as the ratio of the color intersection over the union. The color intersection is defined as the number of voxels for which the position and color match in the two voxel grids, while the union is the total number of voxels occupied within at least one of the voxel grids. The Jaccard index provides a measure of the overlap between the two sets, with a score of 1 indicating a complete match and a score of 0 indicating no match.Sorensen: The Sorensen similarity index is a variant of the Jaccard index that places more weight on the size of the intersection. The Sorensen index is calculated as $\frac{2\cdot |X\cap Y|}{\left|X|+|Y\right|}$, where the numerator is the double of the color intersection and the denominator is the total number of voxels in the compared voxel grids. The Sorensen index provides a measure of the similarity between the two sets, with a score of 1 indicating a complete match and a score of 0 indicating no match.Color Accuracy: The color accuracy (*Color Acc.*) is a metric that measures the similarity between the colors of the voxels in the two compared voxel grids. It is defined as the ratio of the color intersection over the intersection of occupied voxels. This metric provides a measure of the consistency between the colors of the voxels in the two voxel grids.Mean Accuracy: The mean accuracy (*Mean Acc.*) is the mean of the Jaccard, Sorensen, and Color Acc. and serves as the main benchmark measure. This metric provides an overall evaluation of the similarity between the two voxel grids and gives a single, comprehensive score.

## 4. Results

To test the usability of our approach, we assessed our methods using both synthetic data and real-world scenarios. We first present the results obtained from synthetic input generated with the Habitat simulator, followed by the outcomes of real-world tests. Finally, we show results for lower-perspective fine-tuning.

### 4.1. Experimental Setup

The training process for instance segmentation models [21,22] necessitates the incorporation of multiple indoor object datasets. This is crucial because the accuracy of the model on RGB frames and TOF depth estimation directly affects the performance of mesh reconstruction. The primary objective is to acquire a diverse array of samples from various datasets, regardless of the distributions of the individual data sets. This is important, as the final test set will consist of both synthetic and real-world experiments, which will exhibit different distributions. Therefore, the model must be able to generalize effectively.

We selected two datasets [3,28] that contained common household objects relevant to this research. However, the datasets included similar objects under different categories, likely due to the lack of unified annotations provided by crowdsource workers. This necessitated the merging of different classes that referred to the same or similar objects. After the dataset processing, which involved the exclusion of corrupted images, the data was divided into standard training, validation, and test splits. These splits were fixed to ensure consistency across different pre-trained architectures and their backbones.

During label projection, we did not treat the case where an object may not be visible from the top view but becomes visible from a lower viewpoint. The object (a) may be recognizable from this lower viewpoint, or (b) the point cloud may indicate that it is a distinct object. In either case, a new label is to be added, and the label sets will differ for the two views. The label should be “an unknown object” in case (b).

We employed label projection methods with the following parameters: For both the 2D SLIC and the 3D projected SP methods, we set the number of superpixels to 512. In the case of the 3D projected SP, we retained only every fourth point of the point cloud. For the 3D SLIC method, we subsampled the point cloud by setting the voxel size V to 0.1 m and limited the number of supervoxels to 128. Lastly, for the DBSCAN label projection method, a voxel size V of 0.04 m was employed for downsampling, and we set the ϵ parameter of DBSCAN to 0.1, determining the maximum distance between points within a neighborhood. The remaining parameters can be found in the codebase: https://github.com/szilviaszeier/semantic_matching (accessed on 23 May 2023).

We assessed the runtime of each label projection method on a server equipped with an AMD EPYC 7401P CPU and 3 NVIDIA GeForce RTX 2080 Ti GPUs. Despite the optimizations, 3D SLIC and DBSCAN operated at approximately 1 frame per second (FPS). However, the 3D projected SP and the 3D neighborhood methods performed at around 20 and 10 FPS, respectively (see Table 1).

During the lower-perspective fine-tuning, to ensure consistency and minimize training bias, we maintained fixed parameters across all experiments. We set the batch size to 10 due to GPU and data sample size constraints, and trained the model for 50 epochs using the Adam [37] optimizer with a learning rate of 0.0005. During the base model’s fine-tuning process, we performed an extensive search of the parameter space. Based on our experiments, we selected the Transfiner [22] architecture with an R101-FPN-DCN backbone for the pre-trained model, as it yielded the best overall results while maintaining efficient execution speed.

#### 4.1.1. Ground Robot

To record the real-word dataset, we built a small ground robot, shown in Figure 3. We 3D-printed the frame of OpenBot [38], collected and assembled the electrical components from scratch, and adapted the Arduino code to fit our needs. The brain of our agent is the NVIDIA Jetson Nano [39] embedded computer, and we equipped the robot with an Azure Kinect RGBD camera.

#### 4.1.2. Datasets

We conducted experiments in both simulated and real-world environments.

For the simulated dataset, we used the Habitat [12] simulation framework with the Replica [40] dataset, which contains several photo-realistic models of indoor scenes. We selected three scenes: frl_apartment_4, room_2, and office_3, and generated different trajectories for each task. We ran experiments on each scene and reported the average results. We opted for a simulator as it has the added benefit that the ground truth camera poses can be queried, as opposed to an online visual SLAM method.

To create the real-world dataset, we used an Azure Kinect camera to record upper- and lower-view video feeds in two different rooms. Multiple videos were captured from both a human’s and a ground robot’s perspective (see Section 4.1.1). In order to obtain accurate camera poses, we first ran the UcoSLAM algorithm [26] to create a high-quality SLAM map of the environment. This step was important as it allowed for more accurate localization of the camera during recording, resulting in a higher quality reconstruction of the environment [8]. After creating the SLAM map, we used it to localize the camera while recording the area and generating the 3D reconstruction. Since we did not have ground truth semantic annotations of the environment, we manually annotated the resulting mesh for evaluation purposes.

### 4.2. Influence of the Viewpoint

The Habitat simulation framework was used to investigate the effects of unconventional perspectives. For each scene, a trajectory was generated by periodically tilting the camera up and down by 20 degrees. The same trajectory was used for each experiment, but the camera height was varied. Figure 4 shows how the segmentation accuracy is affected by changing the camera height.

Two segmentation models were used, a Mask R-CNN and a Transfiner pre-trained on the SUNRGBD and ADE20K datasets [3,28]. The models performed the worst at 0.2 m height, with mean accuracies of 0.737 and 0.767, while the best results were obtained at 0.8 m height, with mean accuracies of 0.770 and 0.802. Further increases in the height resulted in a slight decay in accuracy.

### 4.3. Three-Dimensional Semantic Reconstruction of Synthetic Scenes

To evaluate the performance of our 3D semantic reconstruction approach on synthetic scenes, we generated a dataset using the Habitat simulation environment. Specifically, we manually controlled the virtual camera along two distinct trajectories within each room, an upper and a lower one, and saved the generated images. In both cases, the aim was to explore the whole room. We ran the 3D semantic reconstruction method from the upper trajectory using different models. Then, we executed the proposed label projection methods given the upper-view reconstruction to generate the lower-view 3D semantic reconstruction. The results are shown in Table 2.

As a baseline, we present the accuracy of the 3D semantic reconstruction using only the semantic segmentation models. We also indicate the accuracy of the label projection methods with the ground truth upper-view reconstruction to demonstrate the upper limit (in terms of the benchmark measures) of the proposed methods.

In all cases, the proposed label projection methods outperformed the baseline methods, i.e., 3D semantic reconstruction using deep learning models applied to the odd viewpoints. The 3D neighborhood method achieved the highest accuracy regardless of which model was used for the upper-view 3D semantic reconstruction. The DBSCAN approach achieved similar color accuracy when used in conjunction with deep learning models, but fell short when using the ground truth mesh due to its inability to detect fine details without thorough fine-tuning. The 3D SLIC and 3D projected SP methods performed similarly. They outperformed DBSCAN with the ground truth and provided comparable results when used with deep learning models.

Figure 5 shows the 3D semantic reconstructions using each approach. The label projection methods were guided by the ground truth upper-view semantic mesh.

### 4.4. Three-Dimensional Semantic Reconstruction of Real Scenes

The experiments performed on the synthetic dataset do not provide a clear picture of the real-world performance of our methods, as the data lacks noise and measurement errors. Therefore, we tested the proposed methods on real-world data as well.

As in the previous section, we recorded images from both lower and upper viewpoints. The upper viewpoint represents the human perspective and was used to generate the 3D semantic reconstruction for guiding the label projection methods. We used the same deep learning models without any weight changes. In Table 3, we present the performance comparison of the 3D semantic reconstruction using the proposed label projection methods.

In contrast to the synthetic dataset, both semantic segmentation models achieved significantly lower scores on the real-world dataset. This can be attributed to the domain shift between the synthetic and real-world data and the challenges posed by the real-world environment. Nonetheless, they obtained similar scores, and the label projection methods improved their performance, particularly with the Transfiner model. Among the label projection methods, the 3D neighborhood approach reached the highest mean accuracy, while DBSCAN achieved the highest color accuracy. However, DBSCAN performed worse compared to other methods, mainly due to its lack of robustness to noise and the diversity of objects in terms of size and overlap. Fine-tuning the parameters of this approach may mitigate these issues, but this would require ground truth samples, which would make the evaluation unfair since other methods do not require supervision.

The 3D SLIC approach achieved a significantly lower mean accuracy, which can be attributed to the measurement errors in the dataset and the steps taken to reduce execution time for real-time runs, such as strong point cloud subsampling and significantly fewer supervoxels. On the other hand, the 3D projected SP method performed better than 3D SLIC, especially when used in combination with the Transfiner model. Notably, the 3D neighborhood approach applied on the Transfiner model achieved results that were comparable to those obtained in the synthetic experiments.

### 4.5. Fine-Tuning on the Lower Perspective

In this section, we explore the potential of fine-tuning the Transfiner [22] model using the label projection results. We begin with identical initial weights and fine-tune the model on three distinct datasets: lower-view ground truth (lower GT), upper-view ground truth segmentations projected onto the robot’s perspective using the 3D neighborhood label projection method (projected GT), and Transfiner predictions projected onto the robot’s perspective using the same label projection method (projected Transfiner). We then evaluate the 3D scenes reconstructed using the fine-tuned models and present the results in Table 4.

To standardize the dataset and address any existing distortions, we apply a series of processing functions. The initial step involves generating polygons for the necessary categories on which the model has been trained, considering that the generated data from the Habitat [12] virtual environment for semantic segmentation is in bitmap format. During this stage, smaller regions are also removed; these areas correspond to segments of larger objects not fully captured in a single frame, and their inclusion would introduce unwanted noise to the dataset (see Figure 6). To achieve this, we independently identify the contours of various categories within a given frame and encapsulate each object in a separate polygon with its associated bounding boxes. Subsequently, we save the generated polygon points in JSON format alongside the corresponding images from which they were derived. This practice helps to reduce the dataset size and maintain uniformity throughout all trials.

Table 4 shows that fine-tuning the Transfiner model on lower-view synthetic data (lower GT) results in higher mean accuracy than fine-tuning the same model on the corresponding lower-view ground truth dataset obtained using the proposed label projection method (projected GT). The introduction of unwanted noise by the label projection method is the likely cause of this difference. Figure 6 displays artifacts introduced by the 3D neighborhood method.

The results also indicate that the proposed superpixel-based edge refinement method has a positive impact on the segmentation accuracy. This is evident from the increase in all benchmark measures for all three datasets. The edge refinement method successfully mitigates the effects of the artifacts introduced by the label projection method, and the models achieve comparable results on the lower GT and projected GT datasets.

Finally, fine-tuning on the projected Transfiner predictions produces a 3D semantic reconstruction with similar accuracy as directly applying the label projection method on the upper-view predictions.

## 5. Conclusions

Several solutions have been proposed in the literature for the 3D semantic reconstruction of indoor environments. However, semantic segmentation in the case of drastic perspective changes has not yet been fully addressed. In this work, we proposed a modular pipeline for semantically reconstructing indoor environments from unusual perspectives, such as one from a small ground robot. Our approach leverages superpixel segmentation and the geometry of the surroundings to extend a partial 3D semantic reconstruction from the human perspective to new, unusual viewpoints.

We experimented in both simulated and real-world scenarios with two different 2D semantic segmentation networks. The proposed label projection methods can provide semantic segmentation from lower viewpoints with accuracy similar to the human perspective. While these methods may introduce artifacts, we proposed a superpixel-based edge refinement step to mitigate their effect. Thereby, label transfer and the fine-tuning of semantic segmentation networks to unusual perspectives becomes possible.

However, while our results are promising, it is important to acknowledge certain limitations inherent to the proposed methods. The resulting 3D reconstruction contains class-level semantic segmentation, which means that we can differentiate between categories but not between the instances themselves. In this work, we need to assume perfect pose estimation and assume a fixed set of classes, as we do not explicitly handle unknown classes. The proposed approaches depend on the partial observation of objects from an upper viewpoint, as the assignment of labels from a lower viewpoint relies on the initial 3D semantic reconstruction from the upper viewpoint. This could pose challenges in scenes where objects are partially obscured. Another potential limitation lies in the segmentation of objects with complex geometries. Due to resolution constraints imposed by the superpixel methods, some of the proposed techniques may have difficulties accurately segmenting such objects.

Despite these limitations, our computationally efficient and real-time capable method opens up new possibilities for human–robot collaboration in various domains, especially in situations where small robots operate in environments designed for human perspectives. These include health and home care applications, where the proposed techniques can potentially enhance robot functionality. In health care settings, it could help robots to navigate in complex environments, provide real-time feedback to patients, and assist in patient care. In home care services, small robots can benefit from a better understanding and interaction with their surroundings, leading to improved assistance in daily tasks. It is important to note, however, that while our method can contribute to these applications, it is but a single component of a larger, more complex system. Overcoming challenges such as handling unknown classes and improving pose estimation accuracy would require significant further research and development. As part of our future work, we aim to integrate our approach with online adaptation methods for segmentation networks, incorporate instance-level segmentation, and facilitate panoptic 3D reconstructions of environments. This would address the aforementioned limitations and further enhance the applicability and accuracy of our approach.

## Figures and Tables

**Figure 1 sensors-23-05126-f001:**
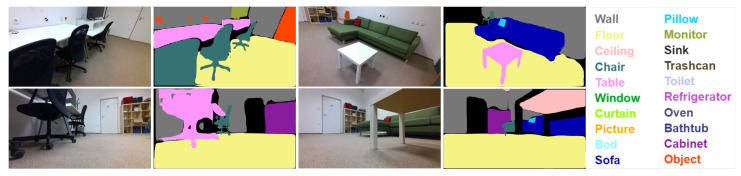
Predicted semantic segmentation from different perspectives. Semantic segmentation algorithms can fail on images taken from unusual perspectives, such as those captured by small ground robots or flying drones (Best viewed in color).

**Figure 2 sensors-23-05126-f002:**
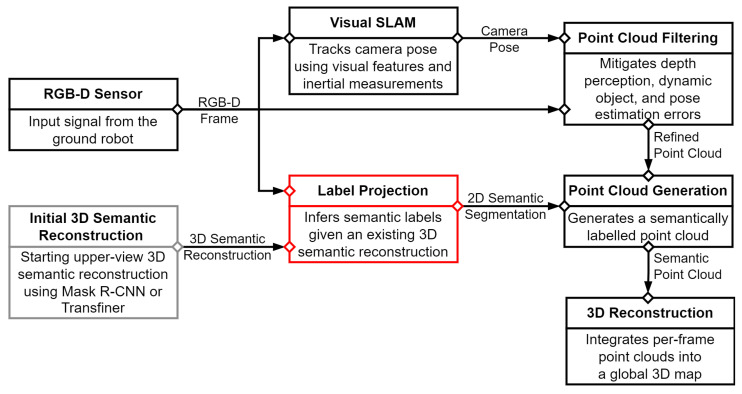
Overall pipeline. We build upon the work of [8]. We added a label projection module (highlighted in red) to infer the missing semantic labels from a lower viewpoint, given an existing 3D semantic reconstruction.

**Figure 3 sensors-23-05126-f003:**
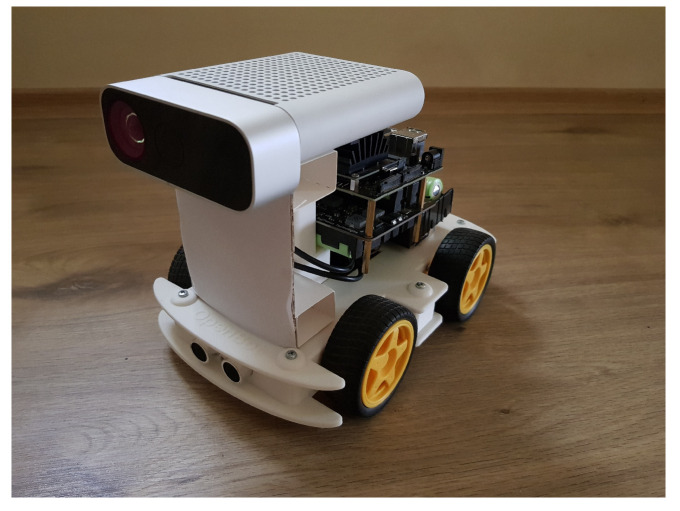
Ground robot used to record the real-world dataset. An OpenBot equipped with an Azure Kinect RGBD camera.

**Figure 4 sensors-23-05126-f004:**
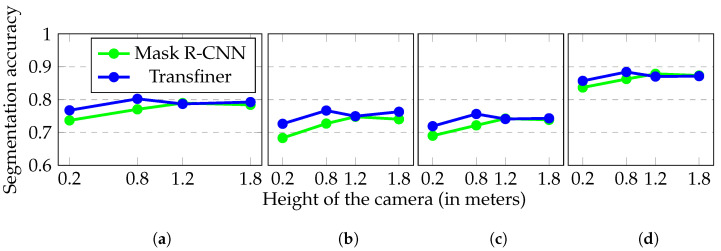
Segmentation accuracy as the height of the camera changes. The same trajectories were used during the evaluations, only the height of the camera was changed. The trajectories were generated by tilting the camera up and down by 20°. Both models were trained on the same dataset. (**a**) Mean Acc. represents the mean of the other three ((**b**) Jaccard, (**c**) Sorensen, and (**d**) Color Acc.) metrics.

**Figure 5 sensors-23-05126-f005:**
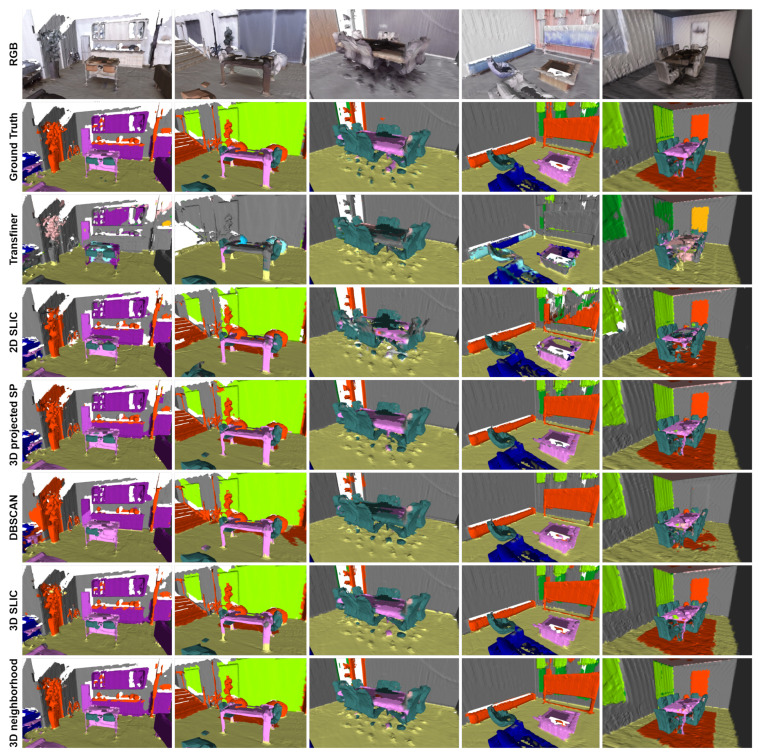
Qualitative results. Each column shows a different environment from the Replica dataset. The label projection methods were guided by the ground truth upper-view 3D semantic reconstructions. We use the same color scheme to represent the labels as in Figure 1 (Best viewed in color).

**Figure 6 sensors-23-05126-f006:**
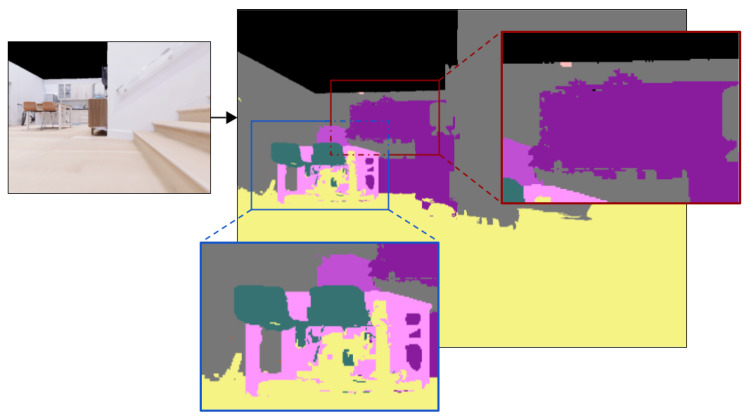
Artifacts introduced by the 3D neighborhood method (Best viewed in color).

**Table 1 sensors-23-05126-t001:** Speed of the 3D label projection methods. The measurements are in frames per second (FPS). For more details, see Section 4.1.

3D SLIC	3D Projected SP	DBSCAN	3D Neighborhood
1.04	9.57	0.84	20.06

**Table 2 sensors-23-05126-t002:** Results on the synthetic dataset. The “view” denotes the camera position. The trajectories differ between the upper and the lower view. From the upper viewpoint, we create a semantic map using per-frame semantic segmentation from different models. (The ground truth model indicates results when the upper-view 3D semantic reconstruction is provided by the simulator.) From the lower viewpoint, we either use a pre-trained model or the upper-view 3D semantic reconstruction with our proposed label projection methods to provide per-frame semantic segmentation to create a lower-view 3D semantic reconstruction. The highest values in each evaluation measure are highlighted for easy reference.

Model	View	Method	Jaccard	Sorensen	Color Acc.	Mean Acc.
Ground Truth	Upper	-	-	-	-	-
	Lower	-	-	-	-	-
		2D SLIC	0.813	0.868	0.955	0.879
		3D SLIC	0.892	0.922	0.975	0.930
		3D projected SP	0.894	0.921	0.969	0.928
		DBSCAN	0.859	0.865	0.953	0.892
		3D neighborhood	**0.917**	**0.944**	**0.985**	**0.949**
Mask R-CNN	Upper	-	0.778	0.754	0.867	0.800
	Lower	-	0.737	0.720	0.841	0.766
		2D SLIC	0.728	0.717	0.852	0.766
		3D SLIC	0.781	0.746	0.861	0.796
		3D projected SP	0.794	0.752	0.860	0.802
		DBSCAN	0.793	**0.757**	**0.871**	0.807
		3D neighborhood	**0.800**	**0.757**	0.865	**0.808**
Transfiner	Upper	-	0.794	0.758	0.867	0.806
	Lower	-	0.734	0.708	0.832	0.758
		2D SLIC	0.718	0.707	0.851	0.759
		3D SLIC	0.790	0.746	0.862	0.799
		3D projected SP	0.798	0.749	0.860	0.802
		DBSCAN	0.796	0.753	**0.871**	0.807
		3D neighborhood	**0.804**	**0.754**	0.866	**0.808**

**Table 3 sensors-23-05126-t003:** Results on the real-world dataset. We used the same methods as described in Table 2. The highest values in each evaluation measure are highlighted for easy reference.

Model	View	Method	Jaccard	Sorensen	Color Acc.	Mean Acc.
Mask R-CNN	Lower	-	0.379	0.522	0.868	0.590
		2D SLIC	0.460	0.604	0.898	0.654
		3D SLIC	0.513	0.644	0.892	0.683
		3D projected SP	0.635	0.727	0.878	0.747
		DBSCAN	0.547	0.680	**0.919**	0.715
		3D neighborhood	**0.665**	**0.749**	0.891	**0.768**
Transfiner	Lower	-	0.384	0.527	0.864	0.592
		2D SLIC	0.509	0.648	0.912	0.690
		3D SLIC	0.565	0.682	0.891	0.713
		3D projected SP	0.697	0.767	0.882	0.782
		DBSCAN	0.540	0.677	**0.927**	0.715
		3D neighborhood	**0.718**	**0.777**	0.881	**0.792**

**Table 4 sensors-23-05126-t004:** Fine-tuning results on the synthetic lower-view dataset. This table shows the results of fine-tuning the Transfiner model on lower-view synthetic data generated using the Habitat simulation framework (lower GT) and the proposed label projection method applied on the upper-view ground truth dataset (projected GT) and on the upper-view Transfiner predictions (projected Transfiner). The edge ref. column indicates whether the superpixel-based edge refinement was applied to the segmentation results before fine-tuning.

Dataset	Edge Ref.	Jaccard	Sorensen	Color Acc.	Mean Acc.
Lower GT	-	0.736	0.801	0.953	0.830
	SLIC 100	0.869	0.802	0.977	0.916
Projected GT	-	0.265	0.393	0.901	0.520
	SLIC 100	0.852	0.892	0.971	0.905
Projected Transfiner	-	0.238	0.347	0.777	0.454
	SLIC 100	0.790	0.753	0.867	0.803

## Data Availability

The repository contains the steps to produce the synthetic data: https://github.com/szilviaszeier/semantic_matching (accessed on 23 May 2023). For access to the real-world data, please get in touch with the corresponding authors.

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
