# Peer review of "Cross-Viewpoint Semantic Mapping: Integrating Human and Robot Perspectives for Improved 3D Semantic Reconstruction"

_sensors, 2023, doi:10.3390/s23115126_

Round 1

Reviewer 1 Report

This paper presents . I have the following observations.

1. Figure 1, predicted semantic segmentation is shown. There are cases where holes are present that lead to flase segmentation results. How do authors deal with such cases?

2. Page-2, line-66, the implementation is said to be at GitHub. After opening the link, it gives 3D Semantic Label Transfer and Matching in Human-Robot Collaboration. Is the implementation of the work presented in the paper under consideration?

3. Page-4, Figure-2, the overall pipeline is too simple which lacks details of the pipeline. It should be revised by including details.

4. Section 3.2.2, Superpixel-based projection in 3D relies on image processing techniques to estimate the depth of objects in a scene. However, this technique can have limitations when the scene contains objects that are partially occluded or have complex geometries.How do authors deal with such issues.

Reviewer 2 Report

This article discusses the use of allocentric semantic 3D maps in human-machine interactions and proposes a method for acquiring semantic labels for images taken from unusual perspectives.

*Strength

- This research aims to address the issue of reduced segmentation performance at a robot's low viewpoint.

- To address the issue, various methods are explored and practical solutions are proposed in the research.

* weakness

- There are not many comparable studies to compare with, as there are not many studies on this topic.

I have a few points below: 

- When describing the viewpoints of humans and robots in the abstract, it would be helpful to specifically mention the viewpoint from the perspective of a small robot. Without the contextual information of a small robot, the difference in viewpoints between humans and robots may not be fully understood.

- It would be helpful to list useful applications where the proposed techniques can be applied in the discussion section.

- (line 133) "... adopted from from previous work [8]." -> "... adopted from previous work [8]."

- It would be helpful to have labels for the x-axis and y-axis in Figure 4.

Round 2

Reviewer 1 Report

The required changes have been made